# Intensive versus less-intensive antileukemic therapy in older adults with acute myeloid leukemia: A systematic review

Yaping Chang[1], Gordon H. Guyatt[1], Trevor Teich[2], Jamie L. Dawdy[3], Shaneela Shahid[1,4], Jessica K. Altman[5], Richard M. Stone[6], Mikkael A. Sekeres[7], Sudipto Mukherjee[7], Thomas W. LeBlanc[8], Gregory A. Abel[9], Christopher S. Hourigan[10], Mark R. Litzow[11], Laura C. Michaelis[12], Shabbir M. H. Alibhai[13], Pinkal Desai[14], Rena Buckstein[15], Janet MacEachern[16], Romina Brignardello-Petersen[1] *

1 Department of Health Research Methods, Evidence, and Impact, McMaster University, Hamilton, Ontario, Canada, 2 Drexel University College of Medicine, Philadelphia, Pennsylvania, United States of America, 3 School of Nursing, McMaster University, Hamilton, Ontario, Canada, 4 Department of Pediatrics, McMaster University, Hamilton, Ontario, Canada, 5 Division of Hematology/Oncology, Feinberg School of Medicine, Northwestern University, Chicago, Illinois, United States of America, 6 Department of Medical Oncology, Dana-Farber Cancer Institute, Boston, Massachusetts, United States of America, 7 Leukemia Program, Cleveland Clinic, Cleveland, Ohio, United States of America, 8 Division of Hematologic Malignancies and Cellular Therapy, Department of Medicine, Duke University School of Medicine, Durham, North Carolina, United States of America, 9 Division of Hematologic Malignances and Population Sciences, Dana-Farber Cancer Institute, Boston, Massachusetts, United States of America, 10 National Heart, Lung, and Blood Institute, National Institutes of Health, Bethesda, Maryland, United States of America, 11 Division of Hematology, Mayo Clinic, Rochester, Minnesota, United States of America, 12 Division of Hematology and Oncology, Department of Medicine, Medical College of Wisconsin, Milwaukee, Wisconsin, United States of America, 13 Department of Medicine, University Health Network & University of Toronto, Toronto, Ontario, Canada, 14 Weill Cornell Medicine, New York City, New York, United States of America, 15 Odette Cancer Centre, Division of Medical Oncology and Hematology, Department of Medicine, Sunnybrook Health Sciences Centre, Toronto, Ontario, Canada, 16 Grand River Regional Cancer Centre, Kitchener, Ontario, Canada

* brignarr@mcmaster.ca

**Data Availability Statement:** All relevant data are within the manuscript and its Supporting information files.

## Abstract

To compare the effectiveness and safety of intensive antileukemic therapy to less-intensive therapy in older adults with acute myeloid leukemia (AML) and intermediate or adverse cyto-genetics, we searched the literature in Medline, Embase, and CENTRAL to identify relevant studies through July 2020. We reported the pooled hazard ratios (HRs), risk ratios (RRs), mean difference (MD) and their 95% confidence intervals (CIs) using random-effects meta-analyses and the certainty of evidence using the GRADE approach. Two randomized trials enrolling 529 patients and 23 observational studies enrolling 7296 patients proved eligible. The most common intensive interventions included cytarabine-based intensive chemotherapy, combination of cytarabine and anthracycline, or daunorubicin/idarubicin, and cytarabine plus idarubicin. The most common less-intensive therapies included low-dose cytarabine alone, or combined with clofarabine, azacitidine, and hypomethylating agent-based chemotherapy. Low certainty evidence suggests that patients who receive intensive versus less-intensive therapy may experience longer survival (HR 0.87; 95% CI, 0.76–0.99), a higher probability of receiving allogeneic hematopoietic stem cell transplantation (RR 6.14; 95% CI, 4.03–9.35), fewer episodes of pneumonia (RR, 0.25; 95% CI, 0.06–

**Funding:** This systematic review was performed as part of the American Society of Hematology (ASH) guidelines for the treatment of older adults with acute myeloid leukemia. The entire guideline development process was funded by ASH. None of the authors received funding specifically for this work.

**Competing interests:** The authors have declared that no competing interests exist.

0.98), but a greater number of severe, treatment-emergent adverse events (RR, 1.34; 95% CI, 1.03–1.75), and a longer duration of intensive care unit hospitalization (MD, 6.84 days longer; 95% CI, 3.44 days longer to 10.24 days longer, very low certainty evidence). Low certainty evidence due to confounding in observational studies suggest superior overall survival without substantial treatment-emergent adverse effect of intensive antileukemic therapy over less-intensive therapy in older adults with AML who are candidates for intensive antileukemic therapy.

## Introduction

Acute myeloid leukemia (AML), the most common type of acute leukemia occurring in adults, presents with a median age of onset of 68 years—more than 75% aged 55 or older [1]—and incurs a 5-year survival of approximately 30% [2,3]. High-risk AML, characterized by advanced patient age, secondary AML, AML with myelodysplastic-related changes or disease carrying adverse cytogenetic or molecular profiles, portends worse survival than disease with favorable or intermediate risk cytogenetic profiles [4,5].

Current standard therapy, typically an intensive chemotherapy (IC) regimen including 3 days of an anthracycline and 7 days of cytarabine (ARA-C), induces remission in 30 to 50% of older patients [6]. Long-term prognosis is, however, poor, with fewer than 10% of individuals over 60 years of age at diagnosis surviving at 5 years post-diagnosis [6–8]. Patients with unfavorable karyotype have minimal or no response to IC and hence an even worse outcome [9]. There are subgroups of AML (e.g., p53 mutated [p53m]) that, regardless of age, have a lower likelihood of responding to IC [10]. For patients with p53m AML, intensive therapy may be inferior to less-intensive therapy [11].

Historically, clinical trials have excluded approximately 40% of older patients on the basis of ineligibility for IC due to comorbidities, age over 75 years, and physician reluctance to aggressively treat older patients [6–9,12].

Azacitidine (AZA), a less-intensive therapy, has also demonstrated efficacy in myelodysplastic syndromes (MDS) and in older patients with AML [12–14]. Subgroup analysis of two prospective randomized trials in older AML patients detected no difference in overall survival (OS) between those treated with AZA or IC [15]. Results from observational studies also suggested that AZA resulted in acceptable median survival times and a survival advantage even in the absence of a complete remission (CR) [16–18]. Therefore, whether AZA or other less-intensive approaches might indeed represent an alternative to IC for the treatment of older patients with AML remains uncertain [19].

The objective of this systematic review was to compare efficacy, safety and quality of life of intensive antileukemic therapy compared to less-intensive antileukemic therapy for patients 55 years and older experiencing newly diagnosed AML with intermediate and adverse cytogenetic or molecular markers and considered appropriate for intensive antileukemic therapy. This systematic review was undertaken to inform the development of the American Society of Hematology (ASH) 2020 Guidelines for Treating Newly Diagnosed Acute Myeloid Leukemia in Older Adults [20].

## Materials and methods

We conducted this systematic review to inform the development of recommendations regarding the treatment of AML in elderly patients from the ASH 2020 Guidelines for Treating

Newly Diagnosed Acute Myeloid Leukemia in Older Adults [20]. As described in detail below, we conducted the study in accordance with the Cochrane Handbook [21] and report the results according to the Preferred Reporting Items for Systematic Reviews and Meta-Analyses guidelines [22].

## Eligibility criteria

**Patients.** We included studies enrolling patients ≥ 55 years of age with newly diagnosed AML including *de novo* AML, treatment-related AML and secondary AML, with adverse- or intermediate-risk cytogenetics and who were considered appropriate for intensive antileukemic therapy. We excluded studies if more than 25% of the patients had one or more of the following characteristics: refractory, recurrent or relapsed AML; acute promyelocytic leukemia, or myeloid conditions related to Down syndrome. We chose 55 years as the age cutoff for our eligibility criterion based on the experts' opinion from ASH guideline panel [20].

**Intervention.** Intensive antileukemic therapy included the following therapies: "7+3" an anthracycline (e.g. daunorubicin, idarubicin, or mitoxantrone) and cytarabine, with or without a third agent (gemtuzumab ozogamicin, vorinostat, bortezomib or midostaurin), with or without hematopoietic growth factor (HGFs, granulocyte colony-stimulating factor [G-CSF], granulocyte-monocyte colony-stimulating factor [GM-CSF], ESAs, or TPO mimetics); FLAG (fludarabine + cytarabine + G-CSF); or CLAG (cladribine + cytarabine + G-CSF). We also included any other antileukemic therapy labelled as intensive by our clinical expert panel (R. M.S, J.K.A. and M.A.S.).

**Comparison.** Less-intensive antileukemic therapy included monotherapy of any one of 5- or 10- day decitabine, gemtuzumab ozogamicin, 5- or 7-day azacitidine, cytarabine that the authors considered "low-dose", clofarabine (if the authors of the study labelled it as a less-intensive therapy), or any of these therapies in combination with other agents. Secondary agents in combinations could include, but were not limited to venetoclax, sorafenib, and HGFs.

**Outcomes.** We included studies in which researchers reported any of the following outcomes: mortality, allogeneic hematopoietic cell transplantation, duration of first morphologic complete remission, severe toxicity, quality of life impairment, functional status impairment, recurrence (or duration of response) and burden on caregivers. We did not address responses less than complete remission, such as partial remission.

**Study designs.** We included randomized controlled trials (RCTs) and comparative observational studies (prospective and retrospective observational studies, before-after studies, and studies in which the comparator was a historical cohort). We excluded studies with less than 10 participants in each arm, and studies published only as conference abstracts.

## Search strategy

For the evidence synthesis supporting the development of recommendations, we searched Medline (via Ovid), Embase (via Ovid), and the Cochrane Central Register of Trials (CENTRAL) from inception to May 2019. For this publication, we updated the search through July 31st, 2020. We conducted an umbrella search encompassing all the questions addressed in the guidelines. We developed structured, database-specific search strategies [23] using terms related to "AML", "chemotherapy" OR "antileukemic therapy", "intensive", "cytarabine", "anthracycline", "idarubicin", "low-intensity treatment", "azacitidine", "decitabine", "aclarubicin" and "LD-AraC", and utilizing Medical Subject Heading (MeSH) terms wherever possible. We included the Medline search strategy as S1 Material in S1 File. We conducted a search of recently completed or ongoing studies using online trial registries (clinicaltrials.gov,

TrialsCentral.org). We further searched the references lists of included studies and previously performed related reviews, and grey literature of dissertations for additional eligible articles.

## Study selection

Pairs of reviewers independently screened titles and abstracts and identified those potentially relevant to this topic. A team of reviewers (Y.C., T.T., J.L.D. and S.S.), working in pairs, screened full texts independently. We conducted calibration exercises before screening and resolved disagreements by discussion and, if necessary, by consulting a third reviewer (R.B.P.).

## Data abstraction and risk of bias assessment

We pilot-tested the data extraction forms, and confirmed in duplicate all abstracted data. To assess the risk of bias for each outcome in each included study, we used the Cochrane Risk of Bias tool 2.0 for RCTs by considering low, unclear, or high risk of bias for domains of random sequence generation, allocation concealment, blinding of patients and personnel, blinding of outcome assessment, incomplete outcome data, selective reporting and other bias [21]. We used the Risk of Bias in Non-randomized studies of interventions (ROBINS-I) for observational studies by considering low, moderate, serious, or critical risk of bias for domains of confounding, selection bias, classification of intervention, deviation from intended interventions, outcome measurement, missing data and selection of reporting result [24]. Reviewers resolved discrepancies through discussion or by a third reviewer when needed (R.B.P.). We collected study and patient demographic information (author, year of publication, country, funding, study design, length of follow-up, sample size, median age, sex distribution, proportion of people with intermediate or adverse cytogenetic, performance status), as well as information regarding each of the treatment arms (regimen, dose, route of administration, cycle) and outcomes of interest. We classified each group as intensive or less-intensive based on eligibility criteria and how the researchers labeled them.

## Effect measures and data analysis

For dichotomous outcomes, we calculated the relative effect of therapies using risk ratios (RRs) and 95% confidence intervals (CIs), which we pooled across studies using random-effects models including the Mantel-Haenszel method [25] and the DerSimonian-Laird estimate of heterogeneity [26]. For continuous outcomes, we used the mean difference (MD) and 95% CI. When a meta-analysis was not possible, we summarized the continuous outcomes by reporting number of intensive- versus less-intensive-therapy comparisons with better and worse outcomes; and by reporting a difference of medians with the method of subtracting the medians from the two arms. For time-to-event outcomes, we used the hazard ratios (HR).

If missing, as is standard, we imputed standard deviations (SD) using median values across similar study characteristics (intervention, follow-up duration) [21]. In order to avoid double counting for studies with more than two treatment arms, we divided the data in the control arm by the number of intervention arms [21]. We performed all analyses using Review Manager 5.3 (The Nordic Cochrane Center, The Cochrane Collaboration, 2014, Copenhagen, Denmark).

## Assessment of certainty of the evidence

We evaluated the certainty of the evidence following the Grading of Recommendations, Assessment, Development and Evaluations (GRADE) approach [27]. According to GRADE, data from randomized controlled trials begin as high certainty evidence but can be rated down

due to moderate, low, or very low due to concerns of risk of bias, imprecision, inconsistency, indirectness, and publication bias [27]. Data from observational studies begin as low certainty of evidence but can be rated down for the same issues as in randomized trials and rated up for large magnitude of effect or dose-response relation [24,27]. We used funnel plots to address publication bias whenever there were 10 or more studies in a meta-analysis. We used GRADE summary of finding tables to present the main findings [28].

### Subgroup and sensitivity analyses

We pooled and reported results from RCTs and observational studies separately.

We prespecified one subgroup analysis: Patients who had intermediate cytogenetic status versus patients who had adverse cytogenetic status, hypothesizing that less-intensive therapy would have larger benefits among patients with intermediate cytogenetic status than among those with adverse cytogenetic profile.

To account for potential reporting bias (i.e. when authors did not report the magnitude of the effect because of lack of statistical significance), we planned a sensitivity analysis for mortality over time. In this analysis, we included studies in which researchers reported that the effect of the therapies was "not statistically significantly different", but did not provide the HR. In the sensitivity analyses we included these studies using a HR of 1 and a CI based on the sample size of the studies.

## Results

### Search results

Following the removal of duplicates, we identified 15615 potential eligible studies of which 231 proved potentially relevant based on title and abstract screening, and 25 studies (7825 patients) proved eligible on full-text review (Fig 1). From the included studies, published between published 2002 and 2020, 21 were included after the first search and informed the development of the recommendations [4,12,14,15,29–45], and 4 were included later [46–49]. We did not find any ongoing studies.

### Study characteristics

Table 1 presents the study characteristics. Two studies (529 patients) were prospective, multi-center RCTs conducted in France, the United Kingdom, Sweden, Italy, Germany, Spain, Australia, the United States, Poland, Belgium, Republic of Korea and Canada [15,29]. Twenty-one were retrospective observational studies (retrospective cohort study, case-control study and case series) (7296 patients) conducted in the United States [4,30–34,47–49], France [33,35–39], the Netherlands [33,40], Republic of Korea [41,42], Japan [43], China [44], Sweden [46], Italy [12,33], Austria, Germany, Portugal and Spain [33]. Two articles reported analyses of data from two trials [14] or three trials [45] in the United States. Since the researchers did not randomize patients for the comparison of interest, we treated the data from these two articles [14,45] as observational studies. Median age of patients in the included studies varied from 63 years to 75 years of age and age range in majority of the studies was between 60 and 90 years.

AML was diagnosed by the World Health Organization (WHO) 2008 criteria (the presence of at least 20% myeloblasts in the bone marrow (BM) or peripheral blood [50]) in 10 studies [4,12,30–32,35–37,42,48], the French-American-British (FAB) criteria (AML was defined by the presence of ≥30% myeloblasts in the marrow or peripheral blood [14,51,52]) in 3 studies [14,38,43], or a combination of WHO and FAB criteria in 3 studies [29,40,44]. In 1

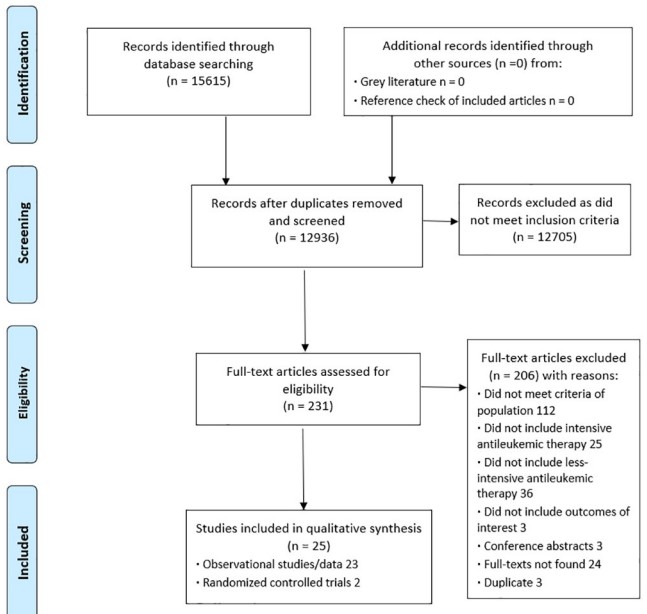

**Fig 1. Eligibility assessment PRISMA flow diagram.**

study > 30% BM blasts was used for the diagnosis of AML [15]. Eight studies did not report criteria used for AML diagnosis [33,34,39,41,45–47,49].

Of the 25 eligible studies, 17 with two-arm parallel comparisons [4,12,31–33,35,37–43,45–48] and 5 from three-arm studies [15,29,34,44,49] provided data suitable for meta-analysis; three articles reported data unsuitable for pooling [14,30,36]. Intensive interventions included cytarabine-based intensive chemotherapy [30,37,40,43], combination of high or intermediate dose of cytarabine and anthracycline [4,29,33–36,38], or daunorubicin/idarubicin [15,32,39,42], FLAG [31,48], IA [14,44,45,47,49], DA [44], MICE [12], or the combinations of intensive chemotherapy agents [41,46]. Less-intensive therapies included LDAC alone [15,29,30,35,43,49], or combined with clofarabine [45], AZA [12,15,29,37,39,40,47], hypomethylating agent (HMA)-based chemotherapy [4,30,32,42,46,48,49], clofarabine [31], decitabine [34,47], gemtuzumab ozogamicin (GO) with or without interleukin (IL)-11 [14], and the various types of less-intensive chemotherapies [33,36,38,41].

## Risk of bias of included studies

We present risk of bias assessments of the observational studies and RCTs in Figs 2 and 3, respectively. Nineteen of the 23 observational studies (82.6%) had moderate to critical risk of bias due to confounding since one or several patient baseline characteristics differed importantly between the treatment groups. Available data indicated that patients in the intensive therapy group were younger in age [30,33–35,37,40,42,46–49], had higher bone marrow blasts (%) [30,37,39,46,47,49], had higher level of white blood cells [30,39,45,46,48,49], or had superior performance status or karyotypic status than patients in the less-intensive therapy group [33,36,37,39,40,42,49]. Ten studies had moderate to serious risk of bias due to deviation from the intended interventions (Fig 2). Of the 2 included RCTs, one had high risk of bias due to problems in random sequence generation and lack of information about allocation concealment [29]; the other had serious high of bias due to lack of blinding of personnel [15] (Fig 3).

**Table 1. Characteristics of included studies.**

| Author (year) | Sample Size | Median age (range, year) | Sex, female, n (%) | People with intermediate or adverse cytogenetics, n (%) | Performance status, tool, n (%) | Intensive antileukemic therapy arm | Less-intensive antileukemic therapy arm | Follow-up duration, median (months) |
|---|---|---|---|---|---|---|---|---|
| Almeida et al. (2017) [32] | 163 | 63 (20–88) | 49 (30.1) | 143 (87.7) | NR | cytarabine-based + daunorubicin/ idarubicin | HMA | 7.7 |
| Boddu et al. (2017) [30] | 802 | 68 (60–75) | NR | 728 (90.8) | ECOG PS Level 0–1, 576 (71.8) Level 2, 131 (16.3) Unknown, 95 (11.9) | cytarabine-based | 1. LDAC; 2. HMA-based | 6.7 |
| Bories et al. (2014) [39] | 210 | 72 (60–89) | 77 (36.7) | 199 (94.8) | Tool NR; PS Level 0–1, 136 (64.8) Level 2–4, 44 (21.0) Unknown, 30 (14.3) | cytarabine-based + daunorubicin/ idarubicin | AZA | 36 |
| Cannas et al. (2015) [38] | 138 | 74 (70–86) | 62 (44.9) | 114 (82.6) | WHO PS >2, 4 (2.9) other categories NR | cytarabine-based + anthracycline | mixed [†] | 13.3 |
| Chen et al. (2016) [44] | 248 | 67 (60–87) | 111 (44.8) | 119 (48.0) | ECOG PS score Level 0 and 1, 85 (34.3) Level 2, 163 (65.7) | 1. IA; 2. DA | CAG | 27.1 |
| Dumas et al. (2017) [37] | 199 | 72 (61–88) | 82 (41.2) | 199 (100) | Tool NR; PS Level 0–1, 123 (61.8) Level 2–3, 49 (24.6) Unknown, 27 (13.6) | cytarabine-based | AZA | 40.8 |
| El-Jawahri et al. (2015) [33] | 330 | 70 (7)* | 135 (40.9) | 305 (92.4) | ECOG PS mean (SD), 0.88 (0.56) | cytarabine-based + anthracycline | mixed [††] | NR (a minimum of 2-year follow-up) |
| Estey et al. (2002) [14] | 82 | 72 (65–89) | NR | 82 (100) | ECOG PS 3 or 4, 11 (13.4) | IA | 1. GO with IL; 2. GO without IL | 4.5 |
| Fattoum et al. (2015) [36] | 183 | 74 (70–86) | 79 (43.2) | 143 (78.1) | WHO PS = < 2, 183 (100) | cytarabine-based + anthracycline | LDAC/AZA/ decitabine | 36 |
| Heiblig et al. (2017) [35] | 195 | 74 (70–86) | 85 (43.6) | 149 (76.4) | WHO PS > = 2, 6 (3.1) | cytarabine-based + anthracycline | LDAC | 36 |
| Maurillo et al. (2018) [12] | 199 | 70 (61–80) | 86 (43.2) | 157 (78.9) | ECOG PS Level 0, 89 (44.7) Level 1, 80 (40.2) Level 2, 30 (15.1) | MICE | AZA | 8.5 |
| Michalski et al. (2019) [34] | 211 | NR (60–69) | 101 (47.9) | 180 (85.3) | 55.9% patients had a KPS score of 90–100; other details NR. | cytarabine-based + anthracycline | 1. mixed; [§] 2. decitabine | NR (reported outcomes at 1-year follow-up) |
| Oh et al. (2017) [42] | 86 | 73 (65–86) | 44 (51.2) | 82 (95.3) | ECOG PS Level 0–1, 59 (68.6) Level 2–4, 25 (29.1) Unknown, 2 (2.3) | cytarabine-based + daunorubicin/ idarubicin | HMA | 20 |
| Osterroos et al. (2020) [46] | 1831 | 71 (60–94) | 812 (44.3) | 1630 (89.0) | WHO PS Level 0, 462 (25.2) Level 1, 968 (52.9) Level 2, 229 (12.5) Level 3, 76 (4.2) Level 4, 31 (1.7) Unknown, 65 (3.5) | IC, unspecified | HMA | 60 |
| Quintas-Cardama et al. (2012) [47] | 671 | 72 (65–89) | 235 (35.0) | 521 (77.6) | ECOG PS Level 0–2, 635 (94.6) | IA | AZA or decitabine | 24 |

(*Continued*)

**Table 1.** (Continued)

| Author (year) | Sample Size | Median age (range, year) | Sex, female, n (%) | People with intermediate or adverse cytogenetics, n (%) | Performance status, tool, n (%) | Intensive antileukemic therapy arm | Less-intensive antileukemic therapy arm | Follow-up duration, median (months) |
|---|---|---|---|---|---|---|---|---|
| Scappaticci et al. (2018) [31] | 64 | 71 (60–83) | NR | 60 (93.8) | NR | FLAG | clofarabine | 20 |
| Solomon et al. (2020) [48] | 262 | 70 (60–88) | 108 (41.2) | 220 (84.0) | NR | FLAG | HMA | 34.2 |
| 1 Takahashi et al. (2016) [45] | 190 | 68 (60–85) | 65 (34.2) | 186 (97.9) | ECOG PS Level 0–1, 161 (84.7) Level 2–3, 29 (15.3) | IA | LDAC + clofarabine | 60 |
| Talati et al. (2020) [49] | 706 | 75 (70–95) | 230 (32.6) | 629 (89.1) | ECOG PS Level 0–1, 593 (84.0) Level 2–4, 99 (14.0) Unknown, 14 (2.0) | IA | 1. HMA; 2. LDAC | 20.5 |
| Tasaki et al. (2014) [43] | 41 | 74 (65–90) | 17 (41.5) | 36 (87.8) | NR | cytarabine-based | LDAC | 9.5 |
| Vachhani et al. (2018) [4] | 201 | 71 (60–93) | 67 (33.3) | 181 (90.0) | NR | cytarabine-based + anthracycline | HMA | 60 |
| van der Helm et al. (2013) [40] | 116 | 67 (60–81) | 52 (44.8) | 109 (94.0) | WHO PS score > = 2, 52 (44.8) | cytarabine-based | AZA | 12 |
| Yi et al. (2014) [41] | 168 | 70 (65–89) | 83 (49.4) | 138 (82.1) | ECOG PS Level 0–1, 68 (40.5) Level 2–4, 100 (59.5) | mixed [§§] | mixed [¶] | 12 |
| Dombret et al.** (2015) [15] | 443 | 75 (64–91) | 184 (41.5) | 440 (99.3) | ECOG PS Level 0–1, 345 (77.9) Level 2, 98 (22.1) | cytarabine-based + daunorubicin/ idarubicin | 1. AZA; 2. LDAC | 24.4 |
| Fenaux et al.** (2009) [29] | 86 | 70 (50–83) | 24 (27.9) | 81 (94.2) | ECOG PS Level 0, 33 (38.4) Level 1, 48 (55.8) Level 2, 4 (4.6) Unknown, 1 (1.2) | cytarabine-based + anthracycline | 1. AZA; 2. LDAC | 20.1 |

* Mean (standard deviation) age.

** Randomized controlled trials.

[†] LDAC(39 patients), AZA (16 patients), decitabine (11 patients), tipifarnib (3 patients), or all-*trans* retinoic acid (ATRA) (1 patient).

[††] Hypomethylating agents, low-dose cytarabine, or single-agent therapy. Single agents included: SNS595 (a topoisomerase II inhibitor), heat-shock protein 90 (HSP90) inhibitor, panobinostat (a histone deacetylase inhibitor), cloretazine, lenalidomide, NEDD-8 activating enzyme inhibitor, sorafenib, PKC-412 inhibitor, and bortezomib.

[§] Five days of decitabine, 5- or 7-day AZA or low-dose cytarabine.

[§§] Anthracycline, high dose cytarabine and fludarabine.

[¶] Low dose cytarabine, hypomethylating agent, arsenic trioxide and all-*trans* retinoic acid (ATRA).

NR, not reported; PS, performance status; ECOG, Eastern Cooperative Oncology Group; WHO, World Health Organization; HMA, hypomethylating agent; LDAC, low-dose cytarabine; AZA, azacitidine; IA, standard-dose cytarabine plus idarubicin; DA, standard-dose cytarabine plus daunorubicin; CAG, cytarabine, aclarubicin, and granulocyte colony-stimulating factor; GO, gemtuzumab ozogamicin; IL, interleukin-11; MICE, mitoxantrone, idarubicin, cytarabine, and etoposide; FLAG, fludarabine, cytarabine, and granulocyte colony-stimulating factor; IC, intensive chemotherapy.

### Relative effects of the interventions

We summarize the effects of the interventions and the certainty of the evidence in GRADE summary of findings tables (Tables 2 and 3).

**All-cause mortality.** *a. Risk of death over time.* Sixteen observational studies (5365 patients) reported hazard ratios (HRs) assessed in a median follow-up time range of 7.7 to 60 months [4,29,31–35,37–40,45–49]. The meta-analysis showed a lower risk of death from any

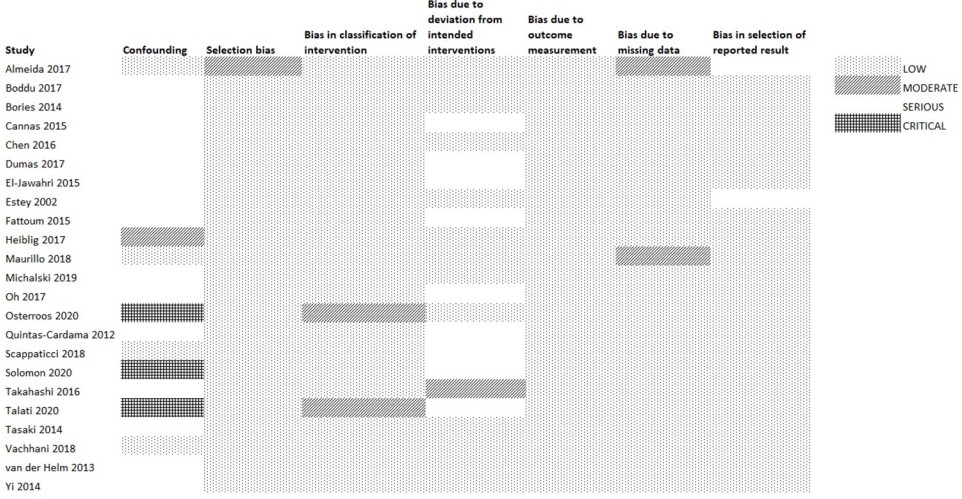

**Fig 2. Risk of bias in observational studies.**

causes with intensive versus less-intensive therapy (HR, 0.87 [95% CI, 0.76–0.99], 50 fewer deaths per 1000, Fig 4, Table 2). We did not detect publication bias for the risk of death over time and presented the funnel plot in Fig 5. The certainty of the evidence was low due to very serious risk of bias.

*b. All-cause mortality at 30 days.* Sixteen observational studies (18 comparisons, 5345 patients) reported all-cause mortality as the proportion of patients who died at 30 days [4,31,32,34,35,37–42,45–49]. The pooled result showed a confidence interval that included a 21% reduction in death and a 92% relative increase (RR, 1.23 [95% CI, 0.79–1.92], S2 Material e-Fig 1 in S2 File, Table 2). The certainty of the evidence was very low due to very serious risk of bias and serious inconsistency.

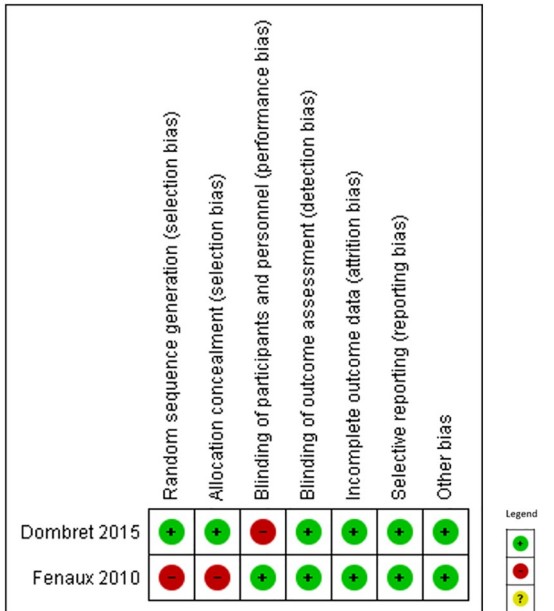

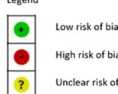

**Fig 3. Risk of bias in RCTs.** RCT, randomized controlled trial.

**Table 2. GRADE summary of findings: Intensive versus less-intensive antileukemic therapy among older patients with acute myeloid leukemia, evidence from observational studies.**

| Outcomes | Relative effects and source of evidence | Absolute effect estimates | | Certainty of evidence | Plain languages summary |
|---|---|---|---|---|---|
| | | Baseline risk for control group (per 1000) | Difference (95% CI) (per 1000) | | |
| Mortality | HR 0.87 (95%CI 0.76 to 0.99) Based on data from 5365 patients in 16 observational studies | 587[1] | -50 (-98 to -4) | Low ⊕⊕◯◯ (Very serious risk of bias)[2] | Intensive antileukemic therapy may reduce mortality. |
| Mortality at 30 days | RR 1.23 (95%CI 0.79 to 1.92) Based on data from 5345 patients in 16 observational studies | 72[3] | 16 (-15 to 66) | Very low ⊕◯◯◯ (Very serious risk of bias and serious inconsistency)[4] | We are very uncertain of the effect of intensive antileukemic therapy on reducing mortality. |
| Mortality at 1 year | RR 0.93 (95%CI 0.85 to 1.02) Based on data from 5724 patients in 18 observational studies | 587[3] | -41 (-88 to 12) | Very low ⊕◯◯◯ (Very serious risk of bias and serious imprecision)[5] | We are very uncertain of the effect of intensive antileukemic therapy on reducing mortality. |
| Allogeneic hematopoietic stem cell transplantation (AlloHCT/AlloSCT) | RR 6.14 (95%CI 4.03 to 9.35) Based on data from 1490 patients in 9 observational studies | 35[3] | 182 (107 to 295) | ⊕⊕⊕◯ Moderate (Very serious risk of bias but strong association)[6] | Intensive antileukemic therapy likely increases AlloHCT/AlloSCT. |
| Serious treatment-emergent adverse events (TEAEs) | RR 1.34 (95%CI 1.03 to 1.75) Based on data from 190 patients in 1 observational study | 463[3] | 157 (14 to 347) | Low ⊕⊕◯◯ (Very serious risk of bias)[2] | Intensive antileukemic therapy may increase TEAEs. |
| Febrile neutropenia (specific TEAE) | RR 1.04 (95%CI 0.93 to 1.15) Based on data from 495 patients in 2 observational studies | 337[3] | 13 (-24 to 51) | Very low ⊕◯◯◯ (Very serious risk of bias and serious imprecision)[5] | We are very uncertain of the effect of intensive antileukemic therapy on febrile neutropenia. |
| Anemia (specific TEAE) | RR 0.75 (95%CI 0.35 to 1.63) Based on data from 431 patients in 1 observational study | 185[3] | -46 (-120 to 117) | Very low ⊕◯◯◯ (Very serious risk of bias and serious imprecision)[5] | We are very uncertain of the effect of intensive antileukemic therapy on anemia. |
| Neutropenia (specific TEAE) | RR 1.30 (95%CI 0.82 to 2.07) Based on data from 431 patients in 1 observational study | 257[3] | -77 (-46 to 275) | Very low ⊕◯◯◯ (Very serious risk of bias and serious imprecision)[5] | We are very uncertain of the effect of intensive antileukemic therapy on neutropenia. |
| Thrombocytopenia (specific TEAE) | RR 0.86 (95%CI 0.47 to 1.56) Based on data from 431 patients in 1 observational study | 252[3] | -35 (-134 to 141) | Very low ⊕◯◯◯ (Very serious risk of bias and serious imprecision)[5] | We are very uncertain of the effect of intensive antileukemic therapy on thrombocytopenia. |
| Pneumonia (specific TEAE) | RR 0.25 (95%CI 0.06 to 0.98) Based on data from 431 patients in 1 observational study | 190[3] | -143 (-179 to -4) | Low ⊕⊕◯◯ (Very serious risk of bias)[2] | Intensive antileukemic therapy may reduce TEAEs. |

*(Continued)*

**Table 2.** (Continued)

| Outcomes | Relative effects and source of evidence | Absolute effect estimates | | Certainty of evidence | Plain languages summary |
|---|---|---|---|---|---|
| | | Baseline risk for control group (per 1000) | Difference (95% CI) (per 1000) | | |
| ICU admission | RR 1.61 (95%CI 0.43 to 6.06) Based on data from 394 patients in 2 observational studies | 176[3] | 107 (-100 to 889) | Low ⊕⊕○○ (Very serious risk of bias)[2] | Intensive antileukemic therapy may increase ICU admission. |

CI, confidence interval; HR, hazard ratio; RR, risk ratio.

[1]We used event rate from 1-year mortality of the less-intensive therapy (from observational study).

[2]Observational studies started at high certainty in the evidence as we used ROBINS-I for assessing risk of bias in individual studies. We have rated down two levels for risk of bias.

[3]We used event rate from the less-intensive therapy to serve as baseline risk.

[4]Observational studies started at high certainty in the evidence as we used ROBINS-I for assessing risk of bias in individual studies. We have rated down two levels for risk of bias. In addition, we rated down for inconsistency (CIs of several studies show minimal or no overlap; $I^2 = 68\%$).

[5]Observational studies started at high certainty in the evidence as we used ROBINS-I for assessing risk of bias in individual studies. We have rated down three levels for risk of bias. In addition, we rated down for imprecision (wide confidence interval includes no difference).

[6]Observational studies started at high certainty in the evidence as we used ROBINS-I for assessing risk of bias in individual studies. We have rated down two levels for risk of bias. The large magnitude of effect (strong association) increased certainty in the evidence.

**Table 3. GRADE summary of findings: Intensive versus less-intensive antileukemic therapy among older patients with acute myeloid leukemia, evidence from RCTs.**

| Outcomes | Relative effects and source of evidence | Absolute effect estimates | | Certainty of evidence | Plain languages summary |
|---|---|---|---|---|---|
| | | Baseline risk for control group (per 1000) | Difference (95% CI) (per 1000) | | |
| Mortality at 1 year | RR 0.90 (95%CI 0.60 to 1.33) Based on data from 87 patients in 1 RCT | 558[1] | -56 (-223 to 184) | Low ⊕⊕○○ (Very serious imprecision)[2] | Intensive antileukemic therapy may reduce mortality. |
| Anemia (specific TEAE) | RR 0.60 (95%CI 0.28 to 1.31) Based on data from 81 patients in 1 RCT | 620[1] | -248 (-446 to 192) | Very low ⊕○○○ (Serious risk of bias and very serious imprecision)[3] | We are very uncertain of the effect of intensive antileukemic therapy on anemia. |
| Neutropenia (specific TEAE) | RR 0.96 (95%CI 0.77 to 1.20) Based on data from 81 patients in 1 RCT | 930[1] | -37 (-214 to 186) | Very low ⊕○○○ (Serious risk of bias and very serious imprecision)[3] | We are very uncertain of the effect of intensive antileukemic therapy on neutropenia. |
| Thrombocytopenia (specific TEAE) | RR 0.94 (95%CI 0.71 to 1.24) Based on data from 81 patients in 1 RCT | 930[1] | -56 (-270 to 223) | Very low ⊕○○○ (Serious risk of bias and very serious imprecision)[3] | We are very uncertain of the effect of intensive antileukemic therapy on thrombocytopenia. |

CI, confidence interval; RR, risk ratio; TEAE, treatment-emergent adverse event.

[1]We used event rate from the less-intensive therapy to serve as baseline risk.

[2]We rated down two levels for imprecision (very wide confidence interval includes important benefit and harm).

[3]We rated down three levels: one for risk of bias (high risk of bias for random sequence generation and allocation concealment), two for imprecision (very wide confidence interval includes important benefit and harm).

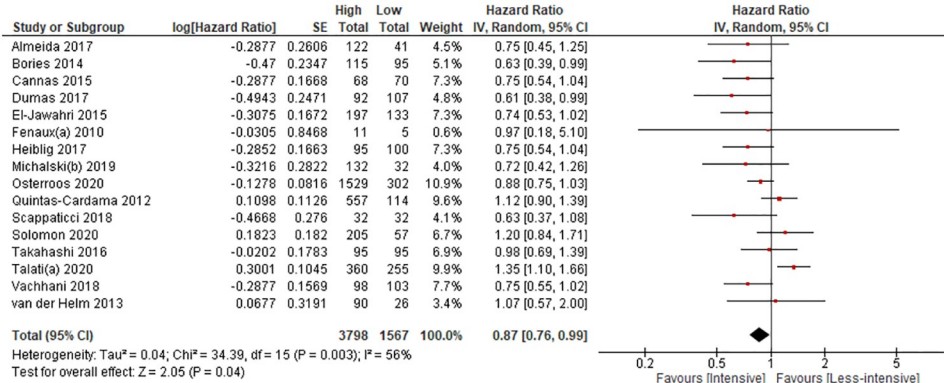

**Fig 4. All-cause mortality assessed with risk of death (all from observational studies).** Intensive, intensive antileukemic therapy; Less-intensive, less-intensive antileukemic therapy; df, degree of freedom; SE, standard error; IV, inverse variance.

*c. All-cause mortality at 1 year*. Eighteen observational studies (21 comparisons, 5724 patients) reported all-cause mortality as the proportion of patients who died at 1 year [4,9,12,31,34,35,38–42,44,46–49]. Results suggested a lower risk of death with intensive therapy over less-intensive therapy (RR, 0.93 [95% CI, 0.85–1.02], S2 Material e-Fig 2 in S2 File, Table 2). The certainty of the evidence was very low due to very serious risk of bias and serious imprecision.

One RCT (87 patients) provided a confidence interval that included a 40% relative reduction in death and a 33% relative increase (RR, 0.90 [95% CI, 0.60–1.33], S2 Material e-Fig 2 in S2 File) [15]. The certainty of the evidence was low due to very serious imprecision.

*d. Overall survival duration*. Pooled estimates were not possible. Eighteen observational studies (22 arm-level comparisons, 6523 patients) reported the median OS duration [4,12,30–37,39,41,42,44–47,49]. Eight reported a shorter overall survival (OS) with intensive therapy compared to less-intensive [30,32,36,37,41,45,49], 13 reported a longer OS with intensive therapy [4,12,31,33–35,39,44,46,47,49], and one reported similar OS durations between the two

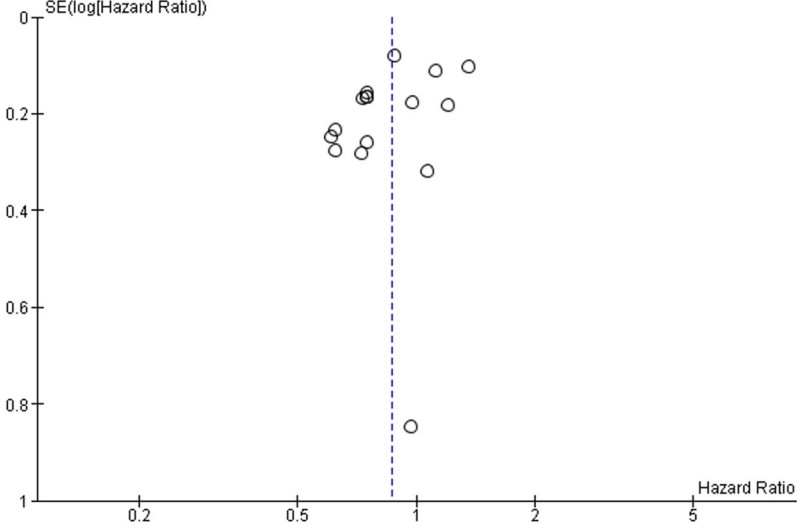

**Fig 5. Funnel plot to detect publication bias.**

groups [42]. The difference in OS duration ranged from 3.6 months shorter to 7.6 months longer when patients received intensive therapy versus less-intensive therapy. Certainty of evidence was very low due to very serious risk of bias, serious inconsistency, and very serious imprecision.

Two RCTs (4 arm-level comparisons, 529 patients) reported the OS duration [15,29]. Three of the four comparisons reported a shorter OS with intensive therapy [29] and one comparison [15] reported a longer OS with intensive therapy. The difference in OS duration ranged from 10.3 months shorter to 5.8 months longer when patients received intensive therapy versus less-intensive therapy. Certainty of evidence was very low due to serious risk of bias, serious inconsistency, and serious imprecision.

**A. Allogeneic hematopoietic (AlloHCT/AlloSCT) stem cell transplantation.** Nine observational studies (10 comparisons, 1490 patients) reported the proportion of people who received AlloHCT/AlloSCT following initial AML therapy [4,12,31,33,34,37,39,40,43]. The meta-analysis showed a higher likelihood of AlloHCT/AlloSCT stem cell transplantation being performed after intensive AML therapy compared to less-intensive therapy (RR, 6.14 [95% CI, 4.03–9.35], 182 more per 1000, S2 Material e-Fig 3 in S2 File, Table 2). The certainty of the evidence was moderate because of strong association in result, though risk of bias was very serious.

**B. Complete remission assessed with time to relapse in months.** Pooled estimates were not possible. Four observational studies (593 patients) reported the time to relapse [4,31,38,45]. Three reported a shorter remission with intensive therapy compared to less-intensive therapy [4,38,45]. The difference in CR duration ranged from 3.1 months shorter to 0.03 months longer when patients received intensive therapy versus less-intensive therapy. One reported similar CR durations between the two groups [31]. The certainty of evidence was very low due to very serious risk of bias, and serious imprecision.

**C. Treatment-emergent adverse events (TEAEs).** *a. Serious TEAEs (Grade 3 to 4 severe toxicity)*. One observational study (190 patients) showed a higher risk of the treatment-emergent Grade 3 to 4 adverse events with intensive therapy over less-intensive therapy at a median follow-up length of 5 years (RR, 1.34 [95% CI, 1.03–1.75], 157 more per 1000, S2 Material e-Fig 4 in S2 File, Table 2) [45]. The certainty of the evidence was low due to very serious risk of bias.

*b. Specific serious TEAEs*. We did not find statistically significant differences between the intensive and less-intensive therapies with respect to the proportion of patients experiencing the specific TEAEs including febrile neutropenia [15,31], anemia [15], neutropenia [29], thrombocytopenia [29] (S2 Material e-Figs 5–8 in S2 File), admission to Intensive Care Unit (ICU) [31,33] (S2 Material e-Fig 10 in S2 File), and duration of hospitalization in days [31,40,45] (S2 Material e-Fig 12 in S2 File), all with low to very low certainty of evidence due to serious imprecision and (or) very serious risk of bias. Tables 2 and 3 present detailed results.

*c. Pneumonia*. One study (RCT that recorded this outcome as a non-randomized manner) (431 patients) showed a lower risk of pneumonia with intensive therapy over less-intensive therapy (RR, 0.25 [95% CI, 0.06–0.98], 143 fewer per 1000, S2 Material e-Fig 9 in S2 File, Table 3) [15]. The certainty of the evidence was low due to very serious risk of bias.

*d. Duration of ICU hospitalization*. Pooled estimates were not possible. One observational study (64 patients) reported a longer ICU hospitalization with intensive therapy over less-intensive therapy (mean difference, 6.84 days longer [95% CI, 3.44 days longer to 10.24 days longer], S2 Material e-Fig 11 in S2 File) [31]. The certainty of the evidence was very low due to very serious risk of bias and very serious imprecision.

**D. Quality of life (QOL) and functional outcomes.** Eligible studies did not report pre-specified outcomes of quality of life impairment, functional status impairment and burden on caregivers.

## Subgroup and sensitivity analyses results

Because the studies did not provide sufficient information to be categorized in subgroups, nor presented outcome data separately according to the cytogenetic status. we did not conduct the preplanned subgroup analysis for patients who had intermediate cytogenetic status versus patients who had adverse cytogenetic status.

For the sensitivity analysis for the outcome risk of death over time, we found 4 observational studies [12,30,42,43] in which researchers reported that the effect of the therapies was not statistically significantly different, but did not provide the HR. We used a HR of 1 and a CI based on sample size and added them to the meta-analysis with the 16 observational studies that reported specific HRs. The meta-analysis of 20 observational studies (6438 patients) showed a lower risk of death with intensive therapy compared to less-intensive therapy (HR, 0.90 [95% CI, 0.82–1.00], S2 Material e-Fig 13 in S2 File), thus not materially different than the initial analysis [4,9,12,29,30,31,33–35,37–40,42,43,45–49]. The certainty of the evidence was low due to very serious risk of bias.

## Discussion

Clinicians and patients considering how aggressively to treat an older adult with AML face a complicated decision. The choice between more or less-intensive chemotherapy is influenced by age, comorbidities, performance status, and most importantly, patient goals of care. Studies to help guide this decision are limited, at times contradictory in their findings, and may be underpowered or prone to bias. Analytic approaches such as meta-analyses can be used to clarify and inform treatment approaches.

Most of the evidence we found comes from observational studies, which resulted in having low certainty evidence due to the high risk of bias owing to confounding: patients who in practice were provided intensive antileukemic therapy are likely to be different from those who were provided less-intensive therapy. This low certainty evidence from observational studies suggests that older patients with newly diagnosed acute myeloid leukemia and with intermediate and adverse cytogenetics who receive intensive antileukemic therapy may be at 23% lower risk of death than those who receive less-intensive antileukemic therapy (Table 2) [4,29,31–35,37–40,45]. Although those who receive more intensive antileukemic therapy are more likely to proceed with stem cell transplant than those who receive less-intensive therapy, the difference may be due to patient and/or disease-related factors influencing the decision regarding initial treatment rather than a higher success rate with intensive chemotherapy, although a higher efficacy (e.g., remission) enabling transplant remains possible.

Because the studies did not provide all data necessary, we were not able to pool results quantifying the difference in survival time between patients who receive intensive versus those who received less intensive antileukemic therapy. Very low certainty evidence reported inconsistent results from both observational studies (shorter survival duration in 7 comparisons [30,32,36,37,41,45] but longer duration in 10 comparisons [4,12,31,33–35,39,44] with intensive therapy; difference ranged from 2.2 months shorter to 7.6 months longer with intensive therapy) and RCTs (shorter survival duration in 3 comparisons [29] and longer duration in 1 comparison [15] with intensive therapy; duration ranged from 10.3 months shorter to 5.8 months longer with intensive therapy). With available data from the included studies, we were not able

to do subgroup analyses for age, cytogenetic status and comorbidities, which might influence the survival durations [6,53].

Low certainty evidence suggests that patients who receive more intensive therapy may be one third more likely (an absolute increase of almost 16%) to experience a grade 3 or worse treatment emergent adverse event, and experience an ICU stay of almost 7 days longer [31], but may be 75% less likely to experience pneumonia (an absolute difference of over 14%) [22] (Table 2). The importance of reduction in pneumonia is unclear in the context of evidence suggesting an increased risk of grade 3 or worse toxicity and prolonged ICU stay.

Our review found almost no data on the impact of different intensities of AML treatment on patient-reported outcomes or functional outcomes such as independence in daily activities. Given the poor long-term survival of many older adults with AML regardless of the intensity of therapy, the impact of treatment intensity on QOL and function represents an important area for further study. Indeed, American Society of Clinical Oncology (ASCO) guidelines [54] recommend that geriatric assessment be employed to identify older adults with cancers such as AML who are at increased risk for poor treatment outcomes, and assessing the effects of intensive versus non-intensive strategies on frailty itself as well as QOL seems a logical next step.

We conducted a rigorous systematic review, using a comprehensive search based on explicit eligibility criteria and multiple independent reviewers for study selection, data abstraction and risk of bias evaluation [21–23]. We applied the GRADE approach to assess the certainty of evidence [27,28], and took additional methodological steps to avoid double counting of studies with multiple treatment arms.

Despite these strengths, due to the nature of the evidence, the certainty of evidence for most outcomes was low to very low based on the non-randomized data; a paucity of randomized data addressed the critical question of whether older patients considered fit for chemotherapy actually have superior outcomes than similar patients receiving less-intensive therapy. Age of 55 years is relatively young, and there were too few data allowing us to dissect out risks of conventionally advanced age (e.g. 70 or 75 years) versus 55–70 or 55–75 years of age in the studies. The evidence includes patients with both intermediate and adverse cytogenetic status. Because of the way in which studies are reported, we could not separate these patients as subgroups and were unable to determine whether treatment would impact differently on the two groups.

For this publication, we updated the original search that informed the development of the recommendations. We included 4 new studies [46–49]. The inclusion of these studies did result in important change in results or certainty of the evidence.

In practice, the physician's assessment of disease, patient characteristics and an analysis of patient goals in the context of anticipated outcomes with each treatment approach are part of the holistic assessment of whether an older adult with AML is considered fit for intensive antileukemic therapy and what is most appropriate induction regimen [53,55].

Intensive antileukemic therapy typically must be delivered in the hospital, representing a burden to the patients and the healthcare system. Intensive chemotherapy, which requires hospitalization due to its effects on myelosuppression and gastrointestinal, may also lead to a longer time in the hospital and greater chance of admission to the ICU [56,57]. However, our review did not find a difference between the two groups for duration of hospitalization ICU hospitalization. Although less-intensive antileukemic therapy can more often be administered in the outpatient setting, it may include more repetitive cycles of therapy than the relatively brief intensive therapy. This ongoing therapy can be difficult for patients to tolerate both psychologically and physically, and may still require hospitalization. The estimates of effect presented in this review, the low certainty of the evidence, and all these considerations resulted in the ASH guideline panel issuing a conditional recommendation for intensive antileukemic therapy over less-intensive antileukemic therapy [20,54].

In conclusion, our results suggest superior overall survival without substantial treatment-emergent adverse effect of intensive antileukemic therapy over less-intensive therapy in older adults with AML who are candidates for intensive antileukemic therapy. The certainty of evidence is almost uniformly low or very low, mainly due to the inherent bias in the selection of intensive chemotherapy for more fit and/or responsive patients in the observational studies that dominated this review. Studies did not address function or QOL [20].

The combination of less-intensive hypomethylating agent therapy with adjunctive agents such as venetoclax therapies [58] targeted against molecular abnormalities such as FLT3 and IDH1/2, and/or the sequencing of less-intensive therapy after initial intensive therapy [59] seem promising and could change the conclusion of similar analyses in the future. Confident resolution of the relative impact of more versus less-intensive chemotherapy for this population will require large, well designed randomized clinical trials reporting subgroup results of patients with varying but prespecified cytogenetic or molecular genetic risks.

## Supporting information

**S1 Checklist.**
(DOCX)

**S1 File. MEDLINE search strategy.** MEDLINE search strategy for intensive versus less-intensive antileukemic therapy in older adults with acute myeloid leukemia.
(DOCX)

**S2 File. Forest plots.** e-Fig 1. All-cause mortality at 30 days after treatment initiation. e-Fig 2. All-cause mortality at 1 year after treatment initiation. e-Fig 3. Proportion of patients who received allogeneic hematopoietic stem cell transplantation. e-Fig 4. Proportion of patients who had serious treatment-emergent adverse events. e-Fig 5. Proportion of patients who had febrile neutropenia. e-Fig 6. Proportion of patients who had anemia. e-Fig 7. Proportion of patients who had neutropenia. e-Fig 8. Proportion of patients who had thrombocytopenia. e-Fig 9. Proportion of patients who had pneumonia. e-Fig 10. Proportion of patients who admitted to intensive care unit (ICU). e-Fig 11. Duration of ICU hospitalization (days). e-Fig 12. Duration of overall hospitalization in days. e-Fig 13. Sensitivity analysis of all-cause mortality assessed with risk of death.
(DOCX)

## Acknowledgments

This systematic review was performed as part of the American Society of Hematology (ASH) guidelines for Treating Newly Diagnosed Acute Myeloid Leukemia in Older Adults. We thank the clinical experts who were part of the panel for critical feedback on population and outcome selection, and treatment grouping: Harry Erba, Hannah Choe, Kristen Odwyer and Ashley Rosko for inputs on the research panel meetings. We thank Shiyun Hu, Thomas Agoritsas, Wojtek Wiercioch, Linn Brandt, Nicolás Yanine for screening full-text studies that were published in non-English languages.

## Author Contributions

**Conceptualization:** Yaping Chang, Gordon H. Guyatt, Jessica K. Altman, Richard M. Stone, Mikkael A. Sekeres, Sudipto Mukherjee, Thomas W. LeBlanc, Gregory A. Abel, Christopher S. Hourigan, Mark R. Litzow, Laura C. Michaelis, Shabbir M. H. Alibhai, Pinkal Desai, Rena Buckstein, Janet MacEachern, Romina Brignardello-Petersen.

**Data curation:** Yaping Chang, Trevor Teich, Jamie L. Dawdy, Shaneela Shahid, Romina Brignardello-Petersen.

**Formal analysis:** Yaping Chang, Gordon H. Guyatt, Trevor Teich, Romina Brignardello-Petersen.

**Investigation:** Gordon H. Guyatt, Jessica K. Altman, Richard M. Stone, Mikkael A. Sekeres, Sudipto Mukherjee, Thomas W. LeBlanc, Gregory A. Abel, Christopher S. Hourigan, Mark R. Litzow, Laura C. Michaelis, Shabbir M. H. Alibhai, Pinkal Desai, Rena Buckstein, Romina Brignardello-Petersen.

**Methodology:** Gordon H. Guyatt, Trevor Teich, Jamie L. Dawdy, Shaneela Shahid, Jessica K. Altman, Richard M. Stone, Mikkael A. Sekeres, Sudipto Mukherjee, Thomas W. LeBlanc, Gregory A. Abel, Christopher S. Hourigan, Mark R. Litzow, Laura C. Michaelis, Shabbir M. H. Alibhai, Pinkal Desai, Rena Buckstein, Janet MacEachern, Romina Brignardello-Petersen.

**Project administration:** Yaping Chang, Gordon H. Guyatt, Romina Brignardello-Petersen.

**Resources:** Gordon H. Guyatt, Romina Brignardello-Petersen.

**Supervision:** Gordon H. Guyatt, Jessica K. Altman, Richard M. Stone, Mikkael A. Sekeres, Sudipto Mukherjee, Thomas W. LeBlanc, Gregory A. Abel, Christopher S. Hourigan, Mark R. Litzow, Laura C. Michaelis, Shabbir M. H. Alibhai, Pinkal Desai, Romina Brignardello-Petersen.

**Validation:** Yaping Chang, Trevor Teich, Jamie L. Dawdy, Shaneela Shahid, Jessica K. Altman, Richard M. Stone, Mikkael A. Sekeres, Sudipto Mukherjee, Thomas W. LeBlanc, Gregory A. Abel, Christopher S. Hourigan, Mark R. Litzow, Laura C. Michaelis, Shabbir M. H. Alibhai, Pinkal Desai, Rena Buckstein, Janet MacEachern, Romina Brignardello-Petersen.

**Writing – original draft:** Yaping Chang, Gordon H. Guyatt, Romina Brignardello-Petersen.

**Writing – review & editing:** Yaping Chang, Gordon H. Guyatt, Trevor Teich, Jamie L. Dawdy, Shaneela Shahid, Jessica K. Altman, Richard M. Stone, Mikkael A. Sekeres, Sudipto Mukherjee, Thomas W. LeBlanc, Gregory A. Abel, Christopher S. Hourigan, Mark R. Litzow, Laura C. Michaelis, Shabbir M. H. Alibhai, Pinkal Desai, Rena Buckstein, Janet MacEachern, Romina Brignardello-Petersen.

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
