## [Decision Letter · Decision Letter 0]

15 Jan 2021

PONE-D-20-39802

Intensive versus less-intensive antileukemic therapy in older adults with acute myeloid leukemia: a systematic review

PLOS ONE

Dear Dr. Brignardello-Petersen,

Thank you for submitting your manuscript to PLOS ONE. After careful consideration, we feel that it has merit but does not fully meet PLOS ONE’s publication criteria as it currently stands. Therefore, we invite you to submit a revised version of the manuscript that addresses the points raised during the review process by Reviewer #2.

We look forward to receiving your revised manuscript.

Kind regards,

Francesco Bertolini, MD, PhD

Academic Editor

PLOS ONE

Journal Requirements:

2. We noticed you have some minor occurrence of overlapping text with the following previous publication by some of the authors of the present study, which needs to be addressed:

- https://ashpublications.org/bloodadvances/article/4/15/3528/461693/American-Society-of-Hematology-2020-guidelines-for

OPTIONAL: The text that needs to be addressed is throughout the discussion.

In your revision ensure you cite all your sources (including your own works), and quote or rephrase any duplicated text outside the methods section.

Further consideration is dependent on these concerns being addressed.

3. We note that your review includes a meta-analysis; please describes any analyses conducted for the assessment of bias across publications using graphical and statistical methods (e.g. funnel plot, Begg/Egger test), or explain the reasons for not doing so.

'..The guidelines development process was funded by ASH and was supported in part by the Intramural Research

Program of the National Heart, Lung, and Blood Institute, National Institutes of Health..'

'The authors received no specific funding for this work.'

6. Please ensure that you refer to Figures 2 and 3 in your text as, if accepted, production will need this reference to link the reader to the figure.

7. Please include captions for your Supporting Information files at the end of your manuscript, and update any in-text citations to match accordingly. Please see our Supporting Information guidelines for more information: http://journals.plos.org/plosone/s/supporting-information

Reviewers' comments:

Reviewer's Responses to Questions

**Comments to the Author**

1. Is the manuscript technically sound, and do the data support the conclusions?

Reviewer #1: Yes

Reviewer #2: Partly

2. Has the statistical analysis been performed appropriately and rigorously? 

Reviewer #1: Yes

Reviewer #2: Yes

3. Have the authors made all data underlying the findings in their manuscript fully available?

Reviewer #1: Yes

Reviewer #2: Yes

4. Is the manuscript presented in an intelligible fashion and written in standard English?

Reviewer #1: Yes

Reviewer #2: Yes

5. Review Comments to the Author

Reviewer #1: The authors present a comprehensive meta-analysis of studies collected through literature in Medline, Embase and CENTRAL to compare the safety and effectiveness of intensive and less-intensive antileukemic therapies in acute myeloid leukemia. The information of the studies and the statistical approaches are extremely well described along the methods and manuscript. The study contains useful information for the readers and the clinical community. In the future it would be great to have a similar analysis accounting for the genetic background of the patients and how it might impact the response to therapies.

I consider this manuscript relevant for clinicians and an excellent guide for the readers. I have no comments.

Reviewer #2: Peer review comments:

Thank you for letting me peer-review this paper. I am a systematic reviewer and have done a couple of reviews on AML therapy, so can appreciate the amount of work that has gone into this review, particularly as the majority of studies are observational.

Overall it is a very well conducted review, using appropriate standard systematic review methods. The writing is clear, however, I think you need to emphasise more strongly the fact that there were only 2 small RCTs and the majority of the results came from observational studies. This is likely to have introduced selection bias particularly as within this disease and within the age groups under study, treatment is often determined by fitness to receive intensive chemotherapy.

Abstract

Reads well, BUT A MAJOR REVISION SHOULD TAKE PLACE REGARDING MORE EMPHASIS ON THE FACT THAT ALL BUT TWO STUDIES ARE OBSERVATIONAL AND THEREFORE THERE WILL BE INHERENT SELECTION BIAS WITHIN THE STUDIES THAT COULD AFFECT THE RESULTS IN FAVOUR OF THE INTENSIVE THERAPY AND THEREFORE THE RESULTS SHOULD BE VIEWED WITH APPROPRIATE CAUTION.

Introduction

A minor comment for the introduction is that you could introduce the origins of the review in that it was undertaken as part of an ASH guideline development, this would then relate to paragraph which talks about it in the discussion. It is always good to see systematic reviews borne from guidelines being published.

Methods

The methods look standard and appear to have been well conducted.

Results

Major comment - there isn’t a separate section for reporting the risk of bias assessment.

It is obvious a lot of work has gone into the risk of bias assessment and I think it is key to understanding the context of the review results. Whether a separate section is inserted or more detail is given within the reporting of the outcome results is a matter for you, but at the moment comments at the bottom of these sections are too vague – saying there is a ‘serious risk of bias’ or ‘series imprecision’ is not telling the reader what the problem is, and the bias could be different depending upon the outcome - especially so with the Robins tool. Many readers, who will hopefully use this in clinical practice, may well be unfamiliar with biases in observational studies and will be unlikely to be able to interpret the Robins tool, so maybe a separate section on the biases might be better. You really need to highlight the main bias at play – looks like its confounding in most of the observational studies – and explain why you are concerned and how this affects the interpretation of the results i.e. cautiously.

Results – minor comment – it would have been helpful if you had put the age ranges as well as the mean age in this table, especially given the review focus is on age.

Results – minor comment – you looked for ongoing trials, did you find any, if so maybe comment on them in the discussion where you talk about future research.

Discussion

The points within Paragraph 2 in the discussion needs more emphasis and the second from last paragraph in the discussion – or something similar - should be in the abstract: ‘The quality of evidence is almost uniformly low or very low mainly due to the inherent bias in the selection of intensive chemotherapy for more fit and/or responsive patients in the observational studies that dominated the review’.

6. PLOS authors have the option to publish the peer review history of their article (what does this mean?). If published, this will include your full peer review and any attached files.

Reviewer #1: No

Reviewer #2: **Yes: **Jayne Wilson

---

## [Author Response · Author response to Decision Letter 0]

23 Feb 2021

Dear Dr. Bertolini and reviewers,

Thank you for taking the time to review our manuscript, and for the positive reviews. Please see below our responses (in red) for the peer review comments regarding our manuscript ID PONE-D-20-39802, “Intensive versus less-intensive antileukemic therapy in older adults with acute myeloid leukemia: a systematic review”. Any changes in the manuscript are tracked. Changes including the additional references were highlighted within the document.

Please let us know if anything requires further clarification.

Sincerely,

Yaping Chang and Romina Brignardello-Petersen

On behalf of the authors

Journal Requirements:

We adjusted the format in manuscript and file names according to PLOS ONE’s style requirements.

2. We noticed you have some minor occurrence of overlapping text with the following previous publication by some of the authors of the present study, which needs to be addressed:

- https://ashpublications.org/bloodadvances/article/4/15/3528/461693/American-Society-of-Hematology-2020-guidelines-for

OPTIONAL: The text that needs to be addressed is throughout the discussion.

In your revision ensure you cite all your sources (including your own works), and quote or rephrase any duplicated text outside the methods section.

We revised the sections and sentences throughout the manuscript, especially in Discussion, to avoid any overlapping text with a previous publication Sekeres et al., 2020. We have also cited the manuscript to make it clear that these clinical considerations come from that work.

3. We note that your review includes a meta-analysis; please describes any analyses conducted for the assessment of bias across publications using graphical and statistical methods (e.g. funnel plot, Begg/Egger test), or explain the reasons for not doing so.

We added a sentence in the methods section describing when we used funnel plots between Lines 365-366 on Page 10.

“We used funnel plots to address publication bias whenever there were 10 or more studies in a meta-analysis.”

We also added a funnel plot to detect publication bias for the outcome of the risk of death as Fig 5 and sentences between Lines 595-596 on Page 22.

“We did not detect publication bias for the risk of death over time and presented the funnel plot in Fig 5.”

'The authors received no specific funding for this work.'

We have removed any funding information in the manuscript.

We would like to update our funding statement as follows “This systematic review was performed as part of the American Society of Hematology (ASH) guidelines for the treatment of older adults with acute myeloid leukemia. The entire guideline development process was funded by ASH. None of the authors received funding specifically for this work”.

5. PLOS requires an ORCID ID for the corresponding author in Editorial Manager on papers submitted after December 6th, 2016. Please ensure that you have an ORCID iD and that it is validated in Editorial Manager.

We added the ORCID ID for the corresponding author Editorial Manager.

6. Please ensure that you refer to Figures 2 and 3 in your text as, if accepted, production will need this reference to link the reader to the figure.

We added the sentence in Results between Lines 551-554 on Page 18.

“Ten studies had moderate to serious risk of bias due to deviation from the intended interventions (Fig 2). Of the 2 included RCTs, one had high risk of bias due to problems in random sequence generation and lack of information about allocation concealment [29]; the other had serious high of bias due to lack of blinding of personnel [15] (Fig 3).”

7. Please include captions for your Supporting Information files at the end of your manuscript, and update any in-text citations to match accordingly.

We added the Supporting Information on Page 42, revised their in-text citations and uploaded the files accordingly.

Reviewer comments: 

Reviewer #1: The authors present a comprehensive meta-analysis of studies collected through literature in Medline, Embase and CENTRAL to compare the safety and effectiveness of intensive and less-intensive antileukemic therapies in acute myeloid leukemia. The information of the studies and the statistical approaches are extremely well described along the methods and manuscript. The study contains useful information for the readers and the clinical community. In the future it would be great to have a similar analysis accounting for the genetic background of the patients and how it might impact the response to therapies.

I consider this manuscript relevant for clinicians and an excellent guide for the readers. I have no comments.

Thank you for your comments.

Reviewer #2: Peer review comments:

Thank you for letting me peer-review this paper. I am a systematic reviewer and have done a couple of reviews on AML therapy, so can appreciate the amount of work that has gone into this review, particularly as the majority of studies are observational.

Overall it is a very well conducted review, using appropriate standard systematic review methods. The writing is clear, however, I think you need to emphasise more strongly the fact that there were only 2 small RCTs and the majority of the results came from observational studies. This is likely to have introduced selection bias particularly as within this disease and within the age groups under study, treatment is often determined by fitness to receive intensive chemotherapy.

Abstract

Reads well, BUT A MAJOR REVISION SHOULD TAKE PLACE REGARDING MORE EMPHASIS ON THE FACT THAT ALL BUT TWO STUDIES ARE OBSERVATIONAL AND THEREFORE THERE WILL BE INHERENT SELECTION BIAS WITHIN THE STUDIES THAT COULD AFFECT THE RESULTS IN FAVOUR OF THE INTENSIVE THERAPY AND THEREFORE THE RESULTS SHOULD BE VIEWED WITH APPROPRIATE CAUTION.

We believe our assessment of certainty of the evidence assessment deals with the problem of study design and potential biases. We agree, however, that this was not explicit for readers, and so we have made this clearer in the discussion. We added a sentence in Abstract between Lines 159-162 on Page 3.

“Low certainty evidence due to confounding in observational studies suggest superior overall survival without substantial treatment-emergent adverse effect of intensive antileukemic therapy over less-intensive therapy in older adults with AML who are candidates for intensive antileukemic therapy.”

Introduction

A minor comment for the introduction is that you could introduce the origins of the review in that it was undertaken as part of an ASH guideline development, this would then relate to paragraph which talks about it in the discussion. It is always good to see systematic reviews borne from guidelines being published.

We added sentences between Lines 230-232 on Page 5.

“This systematic review was undertaken to inform the development of the American Society of Hematology (ASH) 2020 Guidelines for Treating Newly Diagnosed Acute Myeloid Leukemia in Older Adults [20].”

Methods

The methods look standard and appear to have been well conducted.

Thank you for your comment.

Results

Major comment - there isn’t a separate section for reporting the risk of bias assessment.

It is obvious a lot of work has gone into the risk of bias assessment and I think it is key to understanding the context of the review results. Whether a separate section is inserted or more detail is given within the reporting of the outcome results is a matter for you, but at the moment comments at the bottom of these sections are too vague – saying there is a ‘serious risk of bias’ or ‘series imprecision’ is not telling the reader what the problem is, and the bias could be different depending upon the outcome - especially so with the Robins tool. Many readers, who will hopefully use this in clinical practice, may well be unfamiliar with biases in observational studies and will be unlikely to be able to interpret the Robins tool, so maybe a separate section on the biases might be better. You really need to highlight the main bias at play – looks like its confounding in most of the observational studies – and explain why you are concerned and how this affects the interpretation of the results i.e. cautiously.

We added contents in Materials and methods between Lines 318-326 on Page 9.

“To assess the risk of bias for each outcome in each included study, we used the Cochrane Risk of Bias tool 2.0 for RCTs by considering low, unclear, or high risk of bias for domains of random sequence generation, allocation concealment, blinding of patients and personnel, blinding of outcome assessment, incomplete outcome data, selective reporting and other bias [21]. We used the Risk of Bias in Non-randomized studies of interventions (ROBINS-I) for observational studies by considering low, moderate, serious, or critical risk of bias for domains of confounding, selection bias, classification of intervention, deviation from intended interventions, outcome measurement, missing data and selection of reporting result [24].”

We added a separate section in Results between Lines 491-554 on Pages 17-18.

“Risk of bias of included studies

We present risk of bias assessments of the observational studies and RCTs in Fig 2 and Fig 3, respectively. Nineteen of the 23 observational studies (82.6%) had moderate to critical risk of bias due to confounding since one or several patient baseline characteristics differed importantly between the treatment groups. Available data indicated that patients in the intensive therapy group were younger in age [30,33,34,35,37,40,42,46,47,48,49], had higher bone marrow blasts (%) [30,37,39,46,47,49,], had higher level of white blood cells [30,39,45,46,48,49], or had superior performance status or karyotypic status than patients in the less-intensive therapy group [33,36,37,39,40,42,49]. Ten studies had moderate to serious risk of bias due to deviation from the intended interventions (Fig 2). Of the 2 included RCTs, one had high risk of bias due to problems in random sequence generation and lack of information about allocation concealment [29]; the other had serious high of bias due to lack of blinding of personnel [15] (Fig 3).”

We have also added a sentence about this in the Discussion between Lines 756-759 on Page 27.

“Most of the evidence we found comes from observational studies, which resulted in having low certainty evidence due to the high risk of bias owing to confounding: patients who in practice were provided intensive antileukemic therapy are likely to be different from those who were provided less-intensive therapy.”

Results – minor comment – it would have been helpful if you had put the age ranges as well as the mean age in this table, especially given the review focus is on age.

We added age ranges in Table 1 on Pages 14-16 and contents between Lines 448-449 on Page 13.

“Median age of patients in the included studies varied from 63 years to 75 years of age and age range in majority of the studies was between 60 and 90 years.”

Results – minor comment – you looked for ongoing trials, did you find any, if so maybe comment on them in the discussion where you talk about future research.

We added a sentence in Results between Lines 397-398 on Page 12.

“We did not find any ongoing studies.”

Discussion

The points within Paragraph 2 in the discussion needs more emphasis and the second from last paragraph in the discussion – or something similar - should be in the abstract: ‘The quality of evidence is almost uniformly low or very low mainly due to the inherent bias in the selection of intensive chemotherapy for more fit and/or responsive patients in the observational studies that dominated the review’.

We added a sentence in Abstract between Lines 159-162. We added sentences in Discussion between Lines 756-759 on Page 27, and between Lines 836-839, and 849-850 on Page 30.

“Most of the evidence we found comes from observational studies, which resulted in having low certainty evidence due to the high risk of bias owing to confounding: patients who in practice were provided intensive antileukemic therapy are likely to be different from those who were provided less-intensive therapy.”

“In practice, the physician’s assessment of disease, patient characteristics and an analysis of patient goals in the context of anticipated outcomes with each treatment approach are part of the holistic assessment of whether an older adult with AML is considered fit for intensive antileukemic therapy and what is most appropriate induction regimen [53,55].”

---

## [Decision Letter · Decision Letter 1]

11 Mar 2021

Intensive versus less-intensive antileukemic therapy in older adults with acute myeloid leukemia: a systematic review

PONE-D-20-39802R1

Dear Dr. Brignardello-Petersen,

We’re pleased to inform you that your manuscript has been judged scientifically suitable for publication and will be formally accepted for publication once it meets all outstanding technical requirements.

Kind regards,

Francesco Bertolini, MD, PhD

Academic Editor

PLOS ONE

Additional Editor Comments (optional):

Reviewers' comments:

Reviewer's Responses to Questions

**Comments to the Author**

1. If the authors have adequately addressed your comments raised in a previous round of review and you feel that this manuscript is now acceptable for publication, you may indicate that here to bypass the “Comments to the Author” section, enter your conflict of interest statement in the “Confidential to Editor” section, and submit your "Accept" recommendation.

Reviewer #1: All comments have been addressed

Reviewer #2: All comments have been addressed

2. Is the manuscript technically sound, and do the data support the conclusions?

Reviewer #1: Yes

Reviewer #2: Yes

3. Has the statistical analysis been performed appropriately and rigorously? 

Reviewer #1: Yes

Reviewer #2: Yes

4. Have the authors made all data underlying the findings in their manuscript fully available?

Reviewer #1: Yes

Reviewer #2: Yes

5. Is the manuscript presented in an intelligible fashion and written in standard English?

Reviewer #1: Yes

Reviewer #2: Yes

6. Review Comments to the Author

Reviewer #1: (No Response)

Reviewer #2: Good publication, good that it relates to the guidelines that it supports, but offers the science behind the guideline recommendations. This is rarely done even for national and international guidelines as it is seen as 'dual publication' or 'salami slicing' for many journals. Guidelines should have the systematic reviews which support them published in full, otherwise a 'black hole' of evidence exists, that is unhelpful in clinical and patient decision making. Congratulations to the authors and PLOS ONE for - hopefully - getting this published.

7. PLOS authors have the option to publish the peer review history of their article (what does this mean?). If published, this will include your full peer review and any attached files.

Reviewer #1: No

Reviewer #2: **Yes: **Jayne Wilson

---

## [Editor Report · Acceptance letter]

17 Mar 2021

PONE-D-20-39802R1 

Intensive versus less-intensive antileukemic therapy in older adults with acute myeloid leukemia: a systematic review 

Dear Dr. Brignardello-Petersen:

I'm pleased to inform you that your manuscript has been deemed suitable for publication in PLOS ONE. Congratulations! Your manuscript is now with our production department. 

Kind regards, 

on behalf of

Dr. Francesco Bertolini 

Academic Editor

PLOS ONE